# Prevalence and risk predictors of childhood stunting in Bangladesh

**Faruq Abdulla**[1], **Azizur Rahman**[2], **Md. Moyazzem Hossain**[3,4] *

**1** Department of Applied Health and Nutrition, RTM Al-Kabir Technical University, Sylhet, Bangladesh,
**2** School of Computing, Mathematics and Engineering, Charles Sturt University, Wagga Wagga, NSW,
Australia, **3** Department of Statistics, Jahangirnagar University, Savar, Dhaka, Bangladesh, **4** School of
Mathematics, Statistics and Physics, Newcastle University, Newcastle upon Tyne, United Kingdom

* hossainmm@juniv.edu

## Abstract

### Background

The child nutritional status of a country is a potential indicator of socioeconomic development. Child malnutrition is still the leading cause of severe health and welfare problems across Bangladesh. The most prevalent form of child malnutrition, stunting, is a serious public health issue in many low and middle-income countries. This study aimed to investigate the heterogeneous effect of some child, maternal, household, and health-related predictors, along with the quantiles of the conditional distribution of Z-score for height-for-age (HAZ) of under five children in Bangladesh.

### Methods and materials

In this study, a sample of 8,321 children under five years of age was studied from BDHS-2017-18. The chi-square test was mainly used to identify the significant predictors of the HAZ score and sequential quantile regression was used to estimate the heterogeneous effect of the significant predictors at different quantiles of the conditional HAZ distribution.

### Results

The findings revealed that female children were significantly shorter than their male counterparts except at the 75[th] quantile. It was also discovered that children aged 7–47 months were disadvantaged, but children aged 48–59 months were advantaged in terms of height over children aged 6 months or younger. Moreover, children with a higher birth order had significantly lower HAZ scores than 1[st] birth order children. In addition, home delivery, the duration of breastfeeding, and the BCG vaccine and vitamin A received status were found to have varied significant negative associations with the HAZ score. As well, seven or fewer antenatal care visits was negatively associated with the HAZ score, but more than seven antenatal care visits was positively associated with the HAZ score. Additionally, children who lived in urban areas and whose mothers were over 18 years and either normal weight or overweight had a significant height advantage. Furthermore, parental secondary or higher education had a significant positive but varied effect across the conditional HAZ

**Data Availability Statement:** This study is based on the secondary dataset. One can access the data set via the following link http://dhsprogram.com/data/available-datasets.cfm.

**Funding:** The author(s) received no specific funding for this work.

**Competing interests:** The authors have declared that no competing interests exist.

distribution, except for the mother's education, at the 50th quantile. Children from wealthier families were also around 0.30 standard deviations (SD) taller than those from the poorest families. Religion also had a significant relationship with the conditional HAZ distribution in favor of non-Muslim children.

## Conclusions

To enhance children's nutritional levels, intervention measures should be designed considering the estimated heterogeneous effect of the risk factors. This would accelerate the progress towards achieving the targets of Sustainable Development Goals (SDGs) related to child and maternal health in Bangladesh by 2030.

## Introduction

The child nutritional status of a country is a potential indicator of socioeconomic development. Although Bangladesh has tried hard to reduce child malnutrition, it remains the country's leading cause of severe child health and welfare problems. Stunting (i.e., a low linear growth) is the most prevalent form of child malnutrition, repeated infection, and inadequate psychosocial stimulation [1,2], and it is considered a serious public health problem among children in many countries [3]. Therefore, stunting during childhood is the most reliable indication of children's well-being and an accurate marker of societal inequality [1]. Globally, about 150.8 million children aged under five years were stunted in 2017 [4], and nearly 40 percent of stunted children lived in Southern Asia [5]. Bangladesh ranked among the highest rates of child malnutrition in the world, with more than 54% of preschool-aged children suffering from stunting [6]. The rate of stunting among children under five decreased dramatically worldwide, from 47% in 1985 to 21.9% in 2018 [7,8]. In Bangladesh, however, 31% of children under the age of five were stunted in 2017 [7] and 28% in 2019 [9]. The stunted rate among children under the age of five in Bangladesh is still greater than the global rate, notwithstanding a sharp decline in chronic malnutrition as measured by levels of stunting.

Stunting has both short and long-term negative impacts on children's health directly and indirectly, such as a low birth weight, obstructing cognitive development, which affects school achievement, and restricting their life prospects in adulthood [10]. Stunting can also affect a child's social and personal development [11]. The serious consequences of stunting have led to the establishment of worldwide nutrition targets to lessen the prevalence of stunted children under five years of age by 40% before 2025 [12]. This worldwide target has subsequently been supported by the UN Sustainable Development Goal 2 (SDG-2), target 2, with a commitment to end all kinds of malnutrition by the year 2030. Hance, taking any effective different approach to the prevention of stunting could avoid at least 1.7 million childhood deaths [13]. Furthermore, the COVID-19 pandemic has made it more challenging to meet the global nutritional targets for stunting by 2025, especially in low and middle-income countries [14,15]. To reduce the prevalence of stunted children, an effective intervention package must be devised, focusing on the most vulnerable groups.

Many studies have been conducted to identify significant risk factors for stunting in low and middle-income countries, and some have focused on Bangladesh [4,7,10,16–50]. Moreover, several studies have been conducted on child malnourishment in Bangladesh [6,51–54]. However, most of these studies have applied binary or linear regression models to explore the unconditional estimate against the potential factors of child malnutrition. However, if the

relationship between child nutritional status and the various demographic and socioeconomic factors is heterogeneous at the various quantiles of the nutritional distribution, then the adoption of these methods may lead to inappropriate policy intervention measures [55]. In addition, outliers may lead to an overestimation of the effect of the chosen covariates or a loss of information pertinent to intervention and health promotion strategies. Outliers also influenced the estimates of mean and variance of a dataset [56,57], In the presence of outliers, the quantile regression (QR) model provides robust conditional estimates without considering the behavior of the mean and median [58]. It produces estimates that are relatively more unbiased than the estimates generated by the linear regression model when the data violate the assumption of normality [59]. Borooah (2005) applied the QR model to capture the heterogeneity and the determinants of height-for-age in India [60]. Many more studies have also used the QR model for similar purposes [55,61–65]. Therefore, it was crucial to estimate the robust measure of the heterogeneous effect of the significant risk factors of child stunting for designing an effective intervention package to lessen the prevalence of stunting among children in the context of Bangladesh.

In this study, the Z-score of height-for-age (HAZ) was used to measure child growth or stunting but the coefficient of skewness of the target variable showed that the HAZ score does not hold the normality assumption. Therefore, this paper investigates the robust measure of the heterogeneous effect of child, maternal, household, and health-related factors on the stunting status of children aged less than five years in Bangladesh, using the QR model and considering the secondary data collected from the latest Bangladesh Demographic and Health Survey (BDHS-2017-18). The findings of this study will help develop effective intervention strategies to prevent stunting in children and hasten Bangladesh's progress toward achieving the SDGs related to child health status by 2030.

## Methods and materials

### Data and variables

In this study, the secondary data was obtained from a nationally representative survey called the BDHS-2017-18, which is a complete survey that covers the enumeration areas (EAs) of the whole country. Details of the sampling procedure used to conduct the survey are available in the published reports [66]. There was a sample size of 8,334 children under five years of age; however, after cleaning the missing values, the analysis was based on the data from 8,321 children.

The child's nutritional status in the surveyed population is based on the Child Growth Standards recommended by the World Health Organization (WHO). These were constructed using an international sample of culturally, ethnically, and genetically distinct healthy children residing under optimum environments favorable to achieving a child's full genetic growth potential [66]. Among the three main anthropometric indexes for child growth, height-for-age measures linear growth. A child was considered as likely to be stunted if they had a height more than two standard deviations below the reference median of height for that age. Stunting was considered severe if the child's height was more than three standard deviations below the reference median of height for that age [66]. The Z-score for height-for-age (HAZ) was the target variable, and several child's characteristics such as sex, age, duration of breastfeeding, and birth order; maternal attributes such as age, education, and BMI; father's education as well as attributes related to household, community, and health were considered as the explanatory variables in this study. The availability in the BDHS dataset, self-efficacy, and pertinent literature served as the basis for the variable selection.

## Quantile regression

The quantile regression (QR) model was initially introduced by Koenker and Basset in 1978, and nowadays it is extensively applied in various research areas, particularly statistics, econometrics, and public health [55,60–65,67–69]. Suppose $Y$ is a random (response) variable having cumulative distribution function (CDF) $F_Y(y)$, i.e., $F_Y(y) = P(Y \leq y)$ and $X$ is the p-dimensional vector of predictor variables. Then the $\tau$th conditional quantile of $Y$ is described as

$$Q_\tau(Y|X = x) = \{y : F_\tau(y|x)\},$$

where the quantile level $\tau$ varies from 0 to 1.

The QR model portrayed by the conditional $\tau$th quantiles of the response $Y$ for considering the values of predictors $x_1, x_2, \ldots, x_p$ can be expressed as $Q_y(\tau|x_1, x_2, \ldots, x_p) = \beta_0^{(\tau)} + \beta_1^{(\tau)} x_1 + \ldots + \beta_p^{(\tau)} x_p, 0 < \tau < 1$, where $\beta^{(\tau)} = (\beta_0^{(\tau)}, \beta_1^{(\tau)}, \ldots, \beta_k^{(\tau)})^T$ is the unknown vector of parameters.

For a random sample $\{y_1, y_2, \ldots, y_n\}$ of $Y$, it is understood that the sample median minimizes the following sum of absolute deviations, $Median = \arg\min_{\xi \in \mathbb{R}} \sum_{i=1}^{n} \rho_\tau(y_t - \xi)$. Likewise, the general $\tau$th sample quantile $\xi(\tau)$, which is the analog of $Q(\tau)$, is formulated as the minimizer: $\xi(\tau) = \arg\min_{\xi \in \mathbb{R}} \sum_{i=1}^{n} \rho_\tau(y_t - \xi)$, where $\rho_\tau(Z) = Z(\tau - I(Z < 0))$ for $0 < \tau < 1$ denotes the loss function with an indicator function $I(.)$. The loss function $\rho_\tau$ allocates a weight of $\tau$ and $1 - \tau$ for positive residuals $= y_i - \xi$ and negative residuals, respectively. The linear conditional quantile function along with this loss function expands the $\tau$th sample quantile $\xi(\tau)$ to the regression setting in a similar way that the linear conditional mean function expands the sample mean. The OLS estimates is obtained based on the linear conditional mean function $E(Y|X = x) = x'\beta$, by solving $\hat{\beta} = \arg\min_{\beta \in \mathbb{R}^p} \sum_{i=1}^{n} (y_t - x'\beta_i)^2$ [70].

The estimated parameter minimizes the sum of squared residuals as the sample mean minimizes the sum of squares $\mu = \arg\min_{\mu \in \mathbb{R}} \sum_{i=1}^{n} (y_t - \mu)^2$. Quantile regression also estimates the linear conditional quantile function, $(\tau|X = x) = x'\beta(\tau)$, by solving $\hat{\beta}(\tau) = \arg\min_{\beta \in \mathbb{R}^p} \sum_{i=1}^{n} \rho_\tau(y_t - x'\beta_i)^2$. For any quantile, $\tau \in (0,1)$ the quantity $\hat{\beta}(\tau)$ is known as the $\tau$th regression quantile. For example, $\tau = 0.5$, which minimizes the sum of absolute residuals, also corresponds to $L_1$-type or median regression. The set of regression quantiles $\{\beta(\tau): \tau \in (0,1)\}$ is called the quantile process [70].

The QR model aimed at solving the term $\min_{\beta \in \mathbb{R}^p} \left[ \sum_i \tau|e_i| + \sum_i (1-\tau)|e_i| \right]$, where $e_i = y_i - x'_i\beta$ is the $i$th value of unknown errors, $\sum_i \tau|e_i|$ gives the asymmetric penalties $\tau|e_i|$ for over prediction and $\sum_i (1-\tau)|e_i|$ gives the asymmetric penalties $(1-\tau)|e_i|$ for under prediction [70]. The $\tau$th quantile regression estimator $\hat{\beta}(\tau)$ is obtained by minimizing the following objective function over $\beta_\tau$

$$Q(\beta_\tau) = \sum_{i \in \{i: y_i \geq x'_i\beta\}}^{N} \tau|y_i - x'_i\beta_\tau| + \sum_{i \in \{i: y_i < x'_i\beta\}}^{N} (1-\tau)|y_i - x'_i\beta_\tau|$$

where, $0 < \tau < 1$, $i : y_i \geq x'_i\beta$ for over prediction, $i : y_i < x'_i\beta$ for under prediction [70].

**Ethical approval.** Ethical approval was not required as the survey was approved by the local Ethics Committee of Bangladesh and the Ethics Committee of the ICF Macro at Calverton, New York, USA.

## Results

The average of the HAZ score was found to be -1.21 with a standard deviation of 1.22, while the coefficient of skewness and kurtosis were 0.25 and 1.05, respectively. The average HAZ score of less than zero indicates the distribution of the target population's HAZ index had shifted downward, indicating that most of the children were suffering from stunting malnutrition in comparison to the reference population. Moreover, the HAZ score distribution for Bangladeshi under five children was shown to be positively skewed by the coefficient of skewness. The graphical comparison of the HAZ scores against the standard normal variate is shown in Fig 1.

Table 1 shows the prevalence of stunted children among Bangladeshi children aged under five years, along with different characteristics considered in this study. The findings show that the child's stunting prevalence was significantly influenced by their sex, age, birth order, duration of breastfeeding, religion, place of residence and delivery, BCG vaccine and vitamin A uptake, mother's age and BMI, number of ANC visits by mothers during pregnancy, parental education, and household's economic status. It was observed that about 18% of the children were stunted (i.e., HAZ score < -2 SD) and approximately 6% of the children were severely stunted (i.e., HAZ score < -3 SD) in Bangladesh. Around 52% of the children were male but the prevalence rate of stunting for males was relatively less than for female children. Moreover, more than 13% of children were aged less than or equal to 6 months, and the rate of stunting typically increased with the child's age up to two years, after which it decreased slightly. As

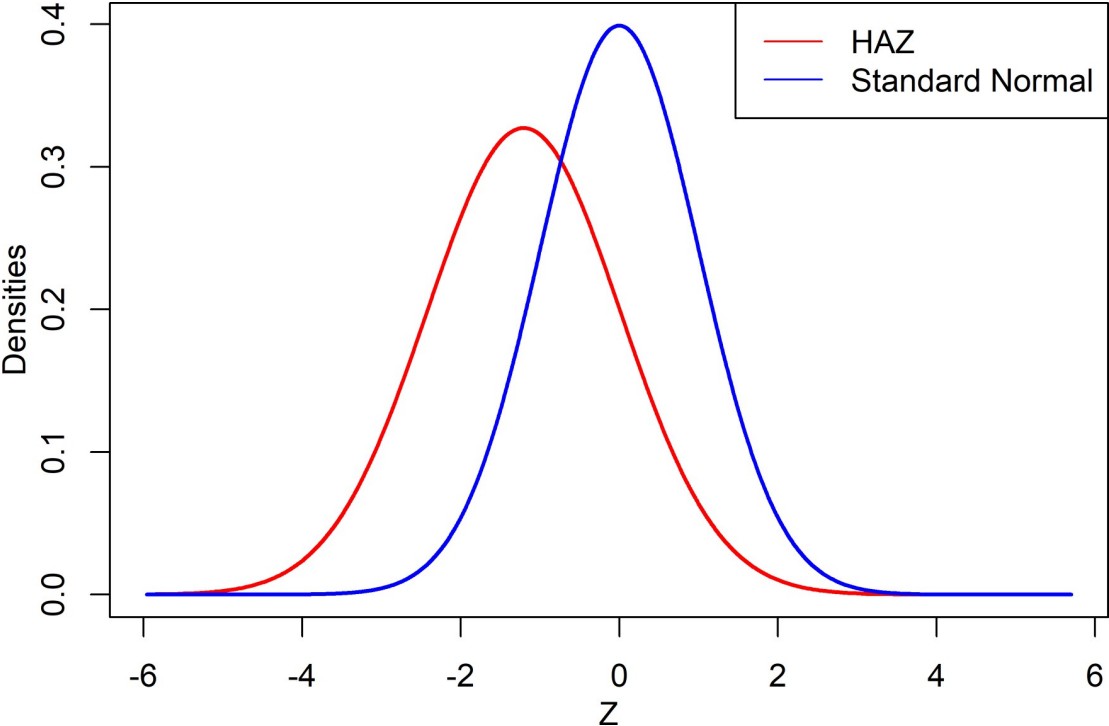

**Fig 1. Density plot of the height–for–age (HAZ) Z–scores and the standard normal variate.**

**Table 1. Prevalence of child categorized according to the anthropometric index height–for–age (stunting) by selected characteristics, Bangladesh (BDHS 2017–18, n = 8,321).**

| Background Characteristics | | Percent (n) | Height-for-Age (Stunting) in % (n) | | P-value of Chi-square |
|---|---|---|---|---|---|
| | | | Severe Stunted (Z<-3 SD) | Moderately Stunted (Z<-2 SD) | |
| *Child Characteristics* | | | | | |
| Sex | Male | 52.16 (4340) | 5.27 (217) | 17.94 (740) | 0.086 |
| | Female | 47.84 (3981) | 6.67 (251) | 18.25 (686) | |
| Child's age (in months) | < = 6 | 13.14 (1093) | 0.87 (9) | 7.93 (83) | <0.001 |
| | 7–12 | 8.2 (683) | 2.67 (18) | 13.24 (88) | |
| | 13–23 | 20.19 (1680) | 8.08 (132) | 23.38 (382) | |
| | 24–35 | 19.87 (1653) | 6.78 (104) | 18.79 (289) | |
| | 36–47 | 19.16 (1594) | 6.6 (98) | 19.75 (292) | |
| | 48–59 | 19.44 (1618) | 7.03 (107) | 19.18 (291) | |
| Birth order | 1st | 38.31 (3188) | 4.92 (148) | 17.68 (532) | <0.001 |
| | 2nd-3rd | 49.22 (4096) | 5.7 (222) | 17.09 (666) | |
| | 4th or higher | 12.46 (1037) | 10 (98) | 23.3 (228) | |
| Duration of breastfeeding | Never breastfeed | 41.28 (3435) | 7.25 (233) | 19.41 (622) | <0.001 |
| | <12 months | 2.05 (171) | 4.62 (8) | 18.84 (31) | |
| | 12 or more | 6.97 (580) | 4.84 (25) | 19.26 (101) | |
| | Still breastfeeding | 49.7 (4135) | 5.07 (202) | 16.84 (671) | |
| Religion | Muslim | 91.96 (7652) | 5.97 (432) | 18.11 (1309) | 0.091 |
| | Non-Muslim | 8.04 (669) | 5.56 (36) | 17.86 (116) | |
| *Parental Characteristics* | | | | | |
| Mother's age (Years) | Up to 18 | 7.23 (601) | 4.55 (26) | 18.42 (107) | <0.001 |
| | 19–24 | 40.24 (3348) | 5.19 (165) | 18.7 (594) | |
| | 25–34 | 44.68 (3718) | 6.68 (235) | 17.43 (615) | |
| | 35 or more | 7.86 (654) | 6.84 (41) | 18.4 (110) | |
| Mother's BMI | Underweight (<18.5) | 13.6 (1132) | 8.66 (96) | 24.81 (274) | <0.001 |
| | Normal (18.5–24.9) | 59.21 (4927) | 5.97 (283) | 18.65 (885) | |
| | Overweight (> = 25) | 27.18 (2262) | 4.37 (89) | 13.11 (266) | |
| Mother's education level | No education | 7.15 (595) | 12.48 (71) | 23.83 (135) | <0.001 |
| | Primary | 28.4 (2363) | 7.92 (179) | 23.06 (521) | |
| | Secondary or above | 64.45 (5363) | 4.32 (218) | 15.22 (769) | |

(*Continued*)

**Table 1.** (Continued)

| Background Characteristics | | Percent (n) | Height-for-Age (Stunting) in % (n) | | |
| --- | --- | --- | --- | --- | --- |
| | | | Severe Stunted (Z<-3 SD) | Moderately Stunted (Z<-2 SD) | P-value of Chi-square |
| *Child Characteristics* | | | | | |
| Father's education level | No education | 14.85 (1218) | 10.88 (126) | 24.77 (286) | <0.001 |
| | Primary | 34.29 (2811) | 7.25 (194) | 21.16 (568) | |
| | Secondary or above | 50.86 (4170) | 3.62 (142) | 13.85 (544) | |
| *Household and Health Characteristics* | | | | | |
| Place of residence | Rural | 73.04 (6078) | 6.23 (361) | 19.56 (1134) | <0.001 |
| | Urban | 26.96 (2243) | 5.12 (107) | 13.99 (292) | |
| Place of delivery | With Health Facility | 49.91 (2551) | 3.74 (91) | 14.31 (347) | <0.001 |
| | Respondent's Home | 50.09 (2561) | 7.01 (173) | 20.12 (495) | |
| Number of antenatal visits during pregnancy | No antenatal visits | 13.13 (644) | 6.61 (40) | 17.14 (105) | 0.001 |
| | 1–3 | 44.66 (2190) | 5.41 (113) | 18.83 (394) | |
| | 4–7 | 36.18 (1774) | 4.33 (74) | 15.71 (268) | |
| | 8 or more | 6.03 (296) | 2.73 (8) | 13.07 (36) | |
| Had diarrhea recently | No | 95.26 (7927) | 5.88 (441) | 18.16 (1362) | 0.107 |
| | Yes | 4.74 (394) | 6.99 (26) | 16.63 (63) | |
| Had fever in last two weeks | No | 66.79 (5558) | 5.62 (294) | 18.16 (949) | 0.325 |
| | Yes | 33.21 (2763) | 6.55 (174) | 17.95 (477) | |
| Had cough in last two weeks | No | 64.01 (5326) | 5.9 (295) | 18.29 (916) | 0.274 |
| | Yes | 35.99 (2995) | 5.99 (172) | 17.73 (510) | |
| Received BCG | No | 6.92 (354) | 1.49 (5) | 8.01 (26) | <0.001 |
| | Yes | 93.08 (4758) | 5.67 (259) | 17.9 (817) | |
| Received vitamin A | No | 30.04 (1536) | 2.84 (42) | 13.23 (194) | <0.001 |
| | Yes | 69.96 (3576) | 6.48 (222) | 18.96 (648) | |
| Wealth index | Poorest | 21.44 (1784) | 8.49 (146) | 24.33 (418) | <0.001 |
| | Poorer | 20.33 (1692) | 7.62 (123) | 21.83 (352) | |
| | Middle | 18.86 (1569) | 4.69 (71) | 18.72 (283) | |
| | Richer | 19.88 (1654) | 5.85 (92) | 15.11 (239) | |
| | Richest | 19.48 (1621) | 2.45 (36) | 9.18 (134) | |
| **Total (Overall)** | | | 5.93 (468) | 18.09 (1425) | |

well, the rate of stunting increased as the child's birth order moved up. Additionally, it was discovered that Muslim children had a slightly higher stunting rate than non-Muslim children (Table 1).

Furthermore, a lower occurrence of childhood stunting was observed among the children whose parents were more educated and/or mothers were either normal or overweight. In addition, stunting prevalence decreased as the number of antenatal visits and/or household's wealth index increased. On the other hand, a higher incidence of stunting was discovered among the children who were born at home and/or lived in rural areas. Additionally, the findings showed that a child's nutritional status was related to their current state of health because stunting was more likely in children with fever or diarrhea than in children without such conditions, even though the association was not significant. Interestingly, children who had received vitamin A and/or the BCG vaccine had a higher rate of stunting (Table 1).

This study considered the 0.10, 0.25, 0.50, 0.75, and 0.90 quantiles. The results of the quantile regression estimation are presented in Table 2, and Fig 2 illustrates the elasticity measured by the 95% confidence interval of the estimated coefficients, which were significant at all quantiles considered. The 95% confidence intervals of all other estimates are presented in S1 Fig. Sequential quantile regression confirmed each of the predictors identified to be significant in bivariate analysis; however, the influence of the significant predictors varied significantly across the conditional distribution of HAZ score. It was revealed that the sex of a child had a significant effect on the HAZ score, except at the 50th quantile. The female children were comparatively shorter than male children except at the 75th quantile, where an inverse scenario was observed. Furthermore, it was shown that children aged 7 to 47 months had a height disadvantage over those aged 6 months or younger, except for those aged 36 to 47 months at the 90th quantile. Those aged 48 to 59 had a height advantage except at the 90th quantile; this may indicate a non-linear relationship between the child's age and HAZ score. Nevertheless, the association's strength and significance were varied throughout the conditional HAZ distribution. Even though the children's 2nd and 3rd birth order showed a significant positive association with the HAZ score at the 10th, 25th, and 75th quantiles, the strength of the association was weak. However, a statistically inverse relationship between the HAZ score and the child's birth order was found for the 2nd and 3rd births at the 90th quantile as well as for the 4th or higher birth order at the 25th and 90th quantiles (Table 2).

Breastfeeding duration was found to have a varied negative association with the HAZ score; however, the association was not significant for a duration of less than 12 months at the 25th and 90th quantiles. BCG vaccine and vitamin A receiving status and the HAZ score were also significantly inversely correlated; however, the strength of the association varied across the conditional HAZ distribution. Although the magnitude of the effect of antenatal care visits increased with the number of visits, the association was not significant at some quantiles. More than seven antenatal care visits were positively associated with the HAZ score, except at the 90th quantile; however, the association was reversed for less than or equal to seven antenatal care visits. In addition, the HAZ score and place of delivery were significantly correlated across all quantiles. The coefficient against home delivery had a negative sign at all quantiles, and its absolute value declined from the lower to the upper quantile. The children who were born at home were comparatively shorter than those born at health facilities, particularly, 0.16 SD and 0.13 SD shorter at 10th and 25th quantiles, respectively. Additionally, except at the 50th quantile, children of mothers older than 18 years exhibited a significant height advantage over those whose mothers were 18 years or younger. Specifically, mothers 35 years or older had children who were 0.45 SD taller than those whose mothers were 18 years or younger at the 90th quantile. At the 90th quantile compared to the 10th quantile, the age groups 19–24, 25–34, and 35+ had 47, 8, and 4 times higher effects on HAZ scores, respectively. There was also a

**Table 2. Quantile regression modeling results of selected risk predictors for stunting (HAZ score) for under five children in Bangladesh, 2017–18.**

| Background Characteristics | Q10 | Q25 | Q50 | Q75 | Q90 |
|---|---|---|---|---|---|
| Sex of child (ref = Male) | | | | | |
| Female | -0.041** | -0.059* | -0.02 | 0.021** | -0.06** |
| Child's age (months) (ref< = 6) | | | | | |
| 7–12 | -0.05 | -0.167* | -0.124* | -0.202 | -0.034 |
| 13–23 | -0.579*** | -0.627*** | -0.592*** | -0.583*** | -0.526** |
| 24–35 | -0.423** | -0.419*** | -0.312*** | -0.24* | -0.184 |
| 36–47 | -0.188** | -0.263** | -0.166** | -0.067 | 0.062 |
| 48–59 | 0.188 | 0.263* | 0.166 | 0.067 | -0.062 |
| Birth order (ref = 1st order) | | | | | |
| 2nd-3rd | 0.008** | 0.011** | 0.05 | 0.032** | -0.046*** |
| 4th or higher | 0.005 | -0.09* | -0.028 | 0.069 | -0.168** |
| Duration of breastfeeding (ref = Never) | | | | | |
| <12 months | -0.728** | -0.873 | -0.576* | -0.118* | -0.481 |
| 12 or more | -0.633** | -0.932* | -0.7** | -0.423*** | -1.069* |
| Still breastfeeding | -0.699** | -0.792* | -0.539* | -0.292** | -1.253* |
| Religion (ref = Muslim) | | | | | |
| Non-Muslim | 0.131* | -0.004 | -0.039 | 0.061** | 0.3** |
| Mother's age (ref = Up to 18 years) | | | | | |
| 19–24 | 0.004 | 0.007** | -0.032*** | -0.005** | 0.189* |
| 25–34 | 0.02** | 0.041* | -0.007* | 0.009** | 0.194** |
| 35 or more | 0.106*** | 0.056 | -0.043*** | 0.038** | 0.454* |
| Mother's education level (ref = No education) | | | | | |
| Primary | 0.155 | 0.052 | -0.057 | -0.079 | -0.063** |
| Secondary or above | 0.294* | 0.171** | -0.016** | 0.047*** | 0.012* |
| Mother's BMI (ref = Underweight (<18.5)) | | | | | |
| Normal (18.5–24.9) | 0.158** | 0.181** | 0.185** | 0.276*** | 0.202 |
| Overweight (> = 25) | 0.162** | 0.255*** | 0.324*** | 0.371*** | 0.34** |
| Father's education level (ref = No education) | | | | | |
| Primary | 0.129 | 0.110* | 0.066 | 0.161** | 0.113 |
| Secondary or above | 0.319*** | 0.346*** | 0.267*** | 0.234** | 0.186* |
| Place of residence (ref = Rural) | | | | | |
| Urban | -0.045 | 0.014* | 0.093** | 0.039* | 0.103 |
| Place of delivery (ref = With health facility) | | | | | |
| Respondent's home | -0.156** | -0.126** | -0.131** | -0.083** | -0.059*** |
| Number of antenatal care visits (ref = None) | | | | | |
| 1–3 | -0.025 | -0.06 | -0.019* | -0.063 | -0.194** |
| 4–7 | -0.022** | -0.056** | 0.031** | -0.024** | -0.087*** |
| 8 or more | 0.068* | 0.069*** | 0.083** | 0.061*** | -0.026* |
| Received BCG (ref = No) | | | | | |
| Yes | -0.384*** | -0.436*** | -0.427*** | -0.535*** | -99.177*** |
| Received vitamin A (ref = No) | | | | | |
| Yes | -0.277*** | -0.159*** | -0.179*** | -0.198** | -0.228* |
| Wealth index (ref = Poorest) | | | | | |
| Poorer | 0.027 | -0.012 | -0.036* | -0.036* | -0.234** |
| Middle | 0.109* | 0.054*** | 0.049* | 0.028** | -0.056*** |
| Richer | 0.198** | 0.139** | 0.058*** | 0.03*** | -0.082*** |
| Richest | 0.31** | 0.314*** | 0.282** | 0.326*** | 0.144** |

(*Continued*)

**Table 2.** (Continued)

| Background Characteristics | Q10 | Q25 | Q50 | Q75 | Q90 |
|---|---|---|---|---|---|
| Intercept | -1.519** | -0.61 | -0.078 | 0.423 | 100.867** |

Notes: p–value<0.01***, 0.01<p–value<0.05**, 0.05<p–value<0.1*.

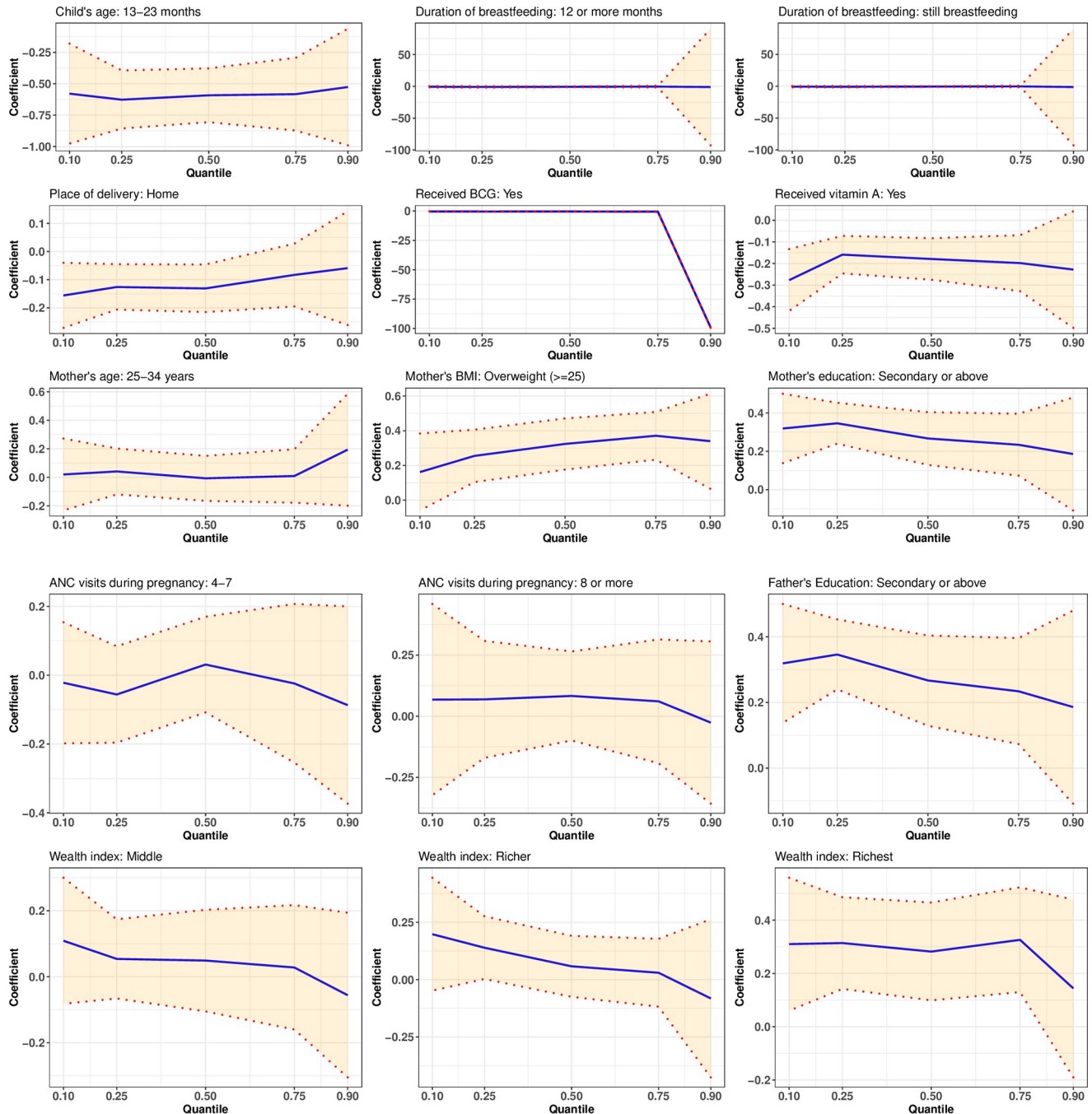

**Fig 2. Elasticity (95% confidence interval) of the significant estimates at all considered quantiles.**

significant positive association between the HAZ score and the mother's BMI, and the extent of the association increased as the mother's BMI increased. The conditional HAZ distribution was largely affected by the mother's BMI at the upper quantiles. Compared to children with underweight mothers, the children whose mothers were normal or overweight were taller. Notably, overweight mothers had children who were 0.32 SD, 0.37 SD, and 0.34 SD taller than those whose mothers were underweight at the 50th, 75th, and 90th quantiles, respectively (Table 2).

Furthermore, parental secondary or above education had a significant but varied effect throughout the conditional HAZ distribution. Compared to children of illiterate parents, children of highly educated parents had higher HAZ scores; however, when the mother's higher education was measured at the 50th quantile, the situation was reversed. Surprisingly, compared to parents who were illiterate, both higher educated fathers and mothers had a positive contribution of around 0.3 SD on their child's HAZ at the 10th quantile; however, the mother's higher education was found to be less important (0.17 SD) than the father's higher education (0.35 SD) for determining the HAZ score at the 25th quantile. The wealth index was also significantly associated with the HAZ score; however, the coefficient for poorer family was insignificant at 10th and 25th quantiles. As household economic status improved, the wealth index's effect increased, becoming more evident at the lower tail of the conditional HAZ distribution. At the 10th, 25th, 50th, and 75th quantiles, children from the richest families were around 0.30 SD taller than those from the poorest families, but the rate decreased to half at the 90th quantile. As well, except at the 10th quantile, the coefficient of urban children was found to have a positive effect on the HAZ score; however, the coefficient was only statistically significant at the 25th, 50th, and 75th quantiles. At the higher quantiles, urban living areas had a greater impact on the HAZ score. The urban children were 0.09 SD taller than their rural counterparts at the 50th quantile. Religion also showed a significant relationship with the conditional HAZ distribution, with the positive sign of the coefficients against non-Muslim children at the 10th, 75th, and 90th quantiles (Table 2).

## Discussion

In this study, the prevalence of stunting and heterogeneous effect of the related risk factors among Bangladeshi children under the age of five were described using a sequential quantile regression model, which is useful to obtain robust estimates of the heterogeneous effect in the presence of outliers or abnormalities in the data [58,59]. There were significant discrepancies in child stunting status by their characteristics like gender, age, birth order, duration of breast-feeding, BCG vaccine and vitamin A receiving status, and place of delivery; maternal characteristics such as age, BMI, education, number of ANC visits during the pregnancy; father's education; household's economic status; and social factors including the type of residence and religion. The child's HAZ score and the predictors had varying degrees of relationship throughout the conditional HAZ distribution. There are several explanations in the literature for this variation [55,62,63,71,72].

The results revealed that female children in Bangladesh had a height disadvantage compared to male children, which is in line with findings from a prior study carried out in Sri Lanka [62] but contrasts with those from many other studies [63,73–77]. This can result from intra-household gender discrimination, especially in food allocation [62]. Therefore, it is important to consider this scenario when developing child nutritional policy initiatives. Moreover, it was discovered that a child's age was negatively associated with their HAZ score up to the age of 47 months, thereafter, an inverse pattern was observed for 48 to 59 months, except at the 90th quantile. Research carried out in Egypt [55], India [63], and Sri Lanka [62] similarly

discovered a negative relationship between a child's age and their HAZ score. It was also found that children with higher birth orders were shorter than those who were their parents' first-born children; however, at some quantiles the opposite was observed. A non-linear relationship between a child's birth order and HAZ score was discovered in India [63], but it was found to be negative in Sri Lanka [62] and many other low and middle-income countries [78]. This conclusion may be supported by the fact that children with higher birth orders may have received less preference in care [79]. Additionally, our results showed a significant inverse relationship between the child's HAZ score and the duration of breastfeeding, BCG vaccination, and vitamin A intake; this result contradicts the findings of earlier studies [63,80–85]. In a previous study, breastfeeding was found to have a non-linear negative correlation by age with the HAZ score [63]. The authors recommend more research concentrating on this issue because this study did not fully explain the reason for this conclusion.

In addition, children whose mothers had received antenatal care more than seven times during their pregnancy had higher HAZ scores, whereas children who were born at home had lower HAZ scores. Therefore, antenatal care and place of delivery were identified as protective factors for child stunting. Earlier studies also identified that antenatal care and place of delivery were related to declines in child stunting [7,55,78,86,87]. By receiving antenatal care throughout pregnancy and giving birth in a health facility, both mother and child can be protected from various health complications such as infections, anemia, iron deficiency, and more [55]. As well, the risk of the child's stunting declines due to the enhanced knowledge and awareness regarding proper diet and health care, and the timely identification of any prevalent health issues learnt by the pregnant women from the health care providers during their antenatal visits and child birth at a health facility [87,88]. A study carried out in India, however, found a non-linear U-shaped association between the number of antenatal visits and HAZ score [63].

Moreover, it was discovered that the mother's age and BMI had a significant positive association with the child's HAZ score, which is in line with the findings of earlier studies [55,86]. However, a different study found that the mother's BMI and the child's HAZ score have a non-linear positive relationship, while the mother's age and the child's HAZ score have an inverse U-shaped relationship [63]. Researchers also found that delaying the age at which mothers have their first child by one year will reduce the rate of stunting by 9% in Ethiopia [86] and 7% in Kenya [89]. Similarly, the rate of stunting will decrease by 4% in Ethiopia [86] and 3% in India [90] for every unit increase in the mother's BMI. The urgency of limiting teenage births is emphasized by the relationship between the mother's BMI and the child's HAZ score [55].

Furthermore, a child's HAZ score was significantly influenced by parental education; children with secondary or higher educated parents had a significant height advantage over those whose parents were illiterate. Many previous studies reported similar findings regarding the relationship between parental education and HAZ score [10,55,62,63,91–96]. It is well documented that parental education is required to improve a child's health, nutrition, and survival since educated parents are more likely to be conscious of health, hygiene, and nutrition issues [55]. Previously, it was reported that a mother's education had a causal nurturing influence on the health of adopted children, as measured by HAZ [55,72]. However, evidence from many countries suggests that knowledge and practices are important pathways, even though the precise mechanism by which parental education influences child outcomes is not well understood [93]. Alongside this, it is widely accepted that better education leads to higher earnings. As a result, higher family earnings allow parents to spend more on health care and good nutrition for their children, which may explain why children of educated parents have a lower risk of stunting [97]. Other related factors like antenatal care visits and a facility based delivery are also influenced by parental education [98,99]. Access to the mass media helps mothers become

more knowledgeable about a wide range of subjects [100]. Therefore, it is recommended to incorporate health and nutrition-related education in mass media programs and the academic educational process in Bangladesh.

Moreover, children of middle-income, more affluent, and the wealthiest families had better HAZ scores than children from the poorest families, which is consistent with the findings of earlier studies [55,63]. This finding is also corroborated by the outcomes from many other studies, indicating that the household economic condition is an important factor in the child's nutritional status in developing countries [10,101–103]. Due to inadequate food consumption, a lack of accessibility to basic health care, and a higher risk of infection, children from lower-income families were more likely to be stunted than children from higher-income families [104–109]. Wealthier families are typically capable of providing better medical treatment, more nutrient-dense food, and an improved and healthier living environment [55,92]. Further, antenatal care and institutional delivery are highly linked to one's wealth [110–112]. Furthermore, disparities in the child's HAZ score by the type of their residence were observed in favor of urban children at upper quantiles of the conditional HAZ distribution. This result aligns with research conducted in Egypt, Jordan, and Yemen [55,113]. The income disparity between urban and rural households could be the reason of this finding [55]. Non-Muslim children were also found to have significantly higher HAZ scores than their Muslim counterparts. A similar finding has been reported in earlier studies [63,80,114]. This finding may be because non-Muslim mothers were more likely to visit a health facility for antenatal care during their pregnancy and give birth in a health facility [114].

## Conclusions

In this study, the authors estimated the robust measure of the heterogeneous influence of some selected child, maternal, household, and health-related characteristics using a QR model along different quantiles of the conditional HAZ distribution of under five children. This study discovered lower HAZ scores in children who were female, under 48 months old, had higher birth orders, were born at home, had mothers who were 18 years old or younger, were underweight, received seven or fewer antenatal care visits, had parents who were illiterate or had less education, were from households with a lower economic background, resided in rural areas, and were Muslims. Therefore, to lessen the burden of stunting in child malnutrition, the authors recommend that the government, NGOs, and community organizations work collaboratively in designing and implementing effective and appropriate nutritional interventions focusing on vulnerable groups, according to the robust findings of this study. The outcomes of this study will help practitioners and policymakers develop and implement robust and cohesive programs to achieve the Sustainable Development Goals associated with child health outcomes in Bangladesh by 2030.

## Supporting information

**S1 Fig. 95% confidence interval of the estimates that were significant at some but not all considered quantiles.**
(TIF)

## Acknowledgments

The authors are grateful to ICF International, Rockville, Maryland, USA, for providing the Bangladesh DHS datasets for this analysis, and to Charles Sturt University for their research

facilities in completing this study. They also thank the academic editor and two reviewers for their valuable comments and suggestions that helped enhance the manuscript's quality.

## Author Contributions

**Conceptualization:** Faruq Abdulla, Azizur Rahman, Md. Moyazzem Hossain.

**Data curation:** Md. Moyazzem Hossain.

**Formal analysis:** Faruq Abdulla, Md. Moyazzem Hossain.

**Methodology:** Azizur Rahman, Md. Moyazzem Hossain.

**Supervision:** Azizur Rahman.

**Visualization:** Faruq Abdulla, Md. Moyazzem Hossain.

**Writing – original draft:** Faruq Abdulla, Md. Moyazzem Hossain.

**Writing – review & editing:** Faruq Abdulla, Azizur Rahman, Md. Moyazzem Hossain.

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
