## [Decision Letter · Decision Letter 0]

4 Sep 2022

PONE-D-22-12168Prevalence and Risk Predictors of Childhood Stunting in BangladeshPLOS ONE

Dear Dr. Hossain,

Thank you for submitting your manuscript to PLOS ONE. After careful consideration, we feel that it has merit but does not fully meet PLOS ONE’s publication criteria as it currently stands. Therefore, we invite you to submit a revised version of the manuscript that addresses the points raised during the review process.

We look forward to receiving your revised manuscript.

Kind regards,

Mohan Kumar

Academic Editor

PLOS ONE

Journal Requirements:

Reviewers' comments:

Reviewer's Responses to Questions

**Comments to the Author**

1. Is the manuscript technically sound, and do the data support the conclusions?

Reviewer #1: Partly

Reviewer #2: Yes

2. Has the statistical analysis been performed appropriately and rigorously? 

Reviewer #1: I Don't Know

Reviewer #2: Yes

3. Have the authors made all data underlying the findings in their manuscript fully available?

Reviewer #1: Yes

Reviewer #2: Yes

4. Is the manuscript presented in an intelligible fashion and written in standard English?

Reviewer #1: No

Reviewer #2: No

5. Review Comments to the Author

Reviewer #1: Dear Authors,

Thank you for the article. The topic is very relevant in the context of the increasing stunting rate in LMIC. However, the manuscript needs major revision, especially in the language aspect. The readers may get confused or lose interest. Kindly consider a professional language editing service before resubmission.

Reviewer #2: At first, I would like congratulate the authors on bringing out a manuscript on this important topic particularly in a n LMIC. However, it needs a few modifications as discussed below.

1. The manuscript needs a thorough language correction.

2. Conclusion needs to be more specific and focused.

3. Discussion needs to be more elaborated with adequate reasoning.

Rest comments have been highlighted in the text

6. PLOS authors have the option to publish the peer review history of their article (what does this mean?). If published, this will include your full peer review and any attached files.

Reviewer #1: No

Reviewer #2: **Yes: **SOUMYA SWAROOP SAHOO

---

## [Author Response · Author response to Decision Letter 0]

26 Sep 2022

Dear Editor,

We would like to sincerely thank the Academic Editor for their valuable comments. We have considered all comments and then thoroughly revised and formatted the manuscript. A detailed response to each particular comment is provided below.

Authors responses to the Editorial Board Member comments:

Thanks. We appreciate your feedback. A careful revision has been conducted to minimize the similarities. 

The required files along with the revised and formatted manuscript are uploaded in the journal system. The revised text are highlighted in “red” color. 

Authors responses to the Journal Requirements:

1. Thank you very much for pointed out it. We have revised the manuscript following the PLOS ONE style. The revised text are highlighted in “red” color. Page: 1-18

2. Thanks. It is not required as the survey was approved by the Ethics Committee of the ICF Macro at Calverton in the USA and by the Ethics Committee in Bangladesh. The revised text are highlighted in “red” color. Page: 7

3. Thanks. We write it in the Methods section. The revised text are highlighted in “red” color. Page: 7

4. Thank you very much. We carefully check the references in the both texts and reference list. 

Authors responses to the Reviewer 1 comments:

Thank you very much for your comments and feedback. We appreciate your feedback. A careful revision has been conducted to fix the grammatical issues. The revised text are highlighted in “red” color. Page: 1-18

Authors responses to the Reviewer 2 comments:

1. Thank you very much for your comments. We appreciate your feedback. A careful revision has been conducted to fix the grammatical issues. 

The revised text are highlighted in “red” color. Page: 1-18

2. Thanks. The Conclusion section is revised as per your comments. The revised text are highlighted in “red” color. Page: 17-18

3. We appreciate your feedback. The Discussion section is revised as per your comments.

The manuscript is also revised as per all highlighted comments appeared in the text. The revised text are highlighted in “red” color. Page: 14-17

In conclusion, the revised version of the manuscript has been produced as per the review outcomes. So, we hope that you would be happy to see this greatly improved version. Once again, we would like to thank you all for your dedication, professional services and cooperation.

Hossain

---

## [Editor Report · Decision Letter 1]

23 Nov 2022

PONE-D-22-12168R1

Prevalence and Risk Predictors of Childhood Stunting in Bangladesh

PLOS ONE

Dear Dr. Hossain,

Thank you for submitting your manuscript to PLOS ONE. After careful consideration, we feel that it has merit but does not fully meet PLOS ONE’s publication criteria as it currently stands. Therefore, we invite you to submit a revised version of the manuscript that addresses the points raised during the review process.

Please note that the previous reviewers have not been invited to assessed the revised mansucript yet. This is because the handling Academic Editor believes that additional copy editing is required for English language presentation before reviewers are Invited.  Please note one of the publication criteria at PLOS ONE (https://journals.plos.org/plosone/s/criteria-for-publication#loc-5 )is that articles must be presented in an intelligible fashion and written in clear, correct, and unambiguous English.  PLOS ONE cannot provide copyediting for manuscripts, as such  we suggest you thoroughly copyedit your manuscript for language usage, spelling, and grammar. Further consideration will be dependent on this aspect of the mansucript.

We look forward to receiving your revised manuscript.

Kind regards,

Lucinda Shen 

Staff Editor 

On behalf of 

Mohan Kumar

Academic Editor

PLOS ONE
---

## [Author Response · Author response to Decision Letter 1]

13 Dec 2022

Dear Dr. Mohan Kumar

Academic Editor

PLOS ONE

We would like to sincerely thank the Academic Editor for their valuable comments. We have considered all comments and then thoroughly revised and formatted the manuscript. A detailed response to each particular comment is provided below.

Author's response to the Academic Editor comments:

1. Thanks. We appreciate your feedback. A careful revision has been conducted to minimize the similarities. 

The required files along with the revised and formatted manuscript are uploaded in the journal system. The revised text are highlighted in “red” color. 

2. Thank you very much. We revised the whole manuscript to fix the grammatical errors. The revised text are highlighted in “red” color.

Page: 1-18

Author's response to the Journal Requirements:

1. Thank you very much for pointed out it. We check the reference list and we ensure that all are complete and correct. 

2. Thanks. We already check all figures using PACE at the time of submit the revised version of the manuscript. 

In conclusion, the revised version of the manuscript has been produced as per the review outcomes. So, we hope that you would be happy to see this greatly improved version. Once again, we would like to thank you all for your dedication, professional services and cooperation.

---

## [Editor Report · Decision Letter 2]

19 Dec 2022

Prevalence and Risk Predictors of Childhood Stunting in Bangladesh

PONE-D-22-12168R2

Dear Dr. Hossain,

We’re pleased to inform you that your manuscript has been judged scientifically suitable for publication and will be formally accepted for publication once it meets all outstanding technical requirements.

Kind regards,

Mohan Kumar

Academic Editor

PLOS ONE

---

## [Editor Report · Acceptance letter]

18 Jan 2023

PONE-D-22-12168R2 

Prevalence and Risk Predictors of Childhood Stunting in Bangladesh 

Dear Dr. Hossain:

I'm pleased to inform you that your manuscript has been deemed suitable for publication in PLOS ONE. Congratulations! Your manuscript is now with our production department. 

Kind regards, 

on behalf of

Dr. Mohan Kumar 

Academic Editor

PLOS ONE